# Two Pediatric Cases of Multisystem Inflammatory Syndrome with Overlapping Neurological Involvement Following SARS-CoV-2 Vaccination and Unknown SARS-CoV2 Infection: The Importance of Pre-Vaccination History

**DOI:** 10.3390/vaccines10071136

**Published:** 2022-07-16

**Authors:** Veronica Santilli, Emma Concetta Manno, Carmela Giancotta, Chiara Rossetti, Nicola Cotugno, Donato Amodio, Gioacchino Andrea Rotulo, Annalisa Deodati, Roberto Bianchi, Giulia Lucignani, Daniela Longo, Massimiliano Valeriani, Paolo Palma

**Affiliations:** 1Academic Department of Pediatrics (DPUO), Research Unit of Clinical Immunology and Vaccinology, Bambino Gesù Children’s Hospital, IRCCS, 00165 Rome, Italy; veronica.santilli@opbg.net (V.S.); emma.manno@opbg.net (E.C.M.); carmela.giancotta@opbg.net (C.G.); nicola.cotugno@opbg.net (N.C.); donato.amodio@opbg.net (D.A.); 2Chair of Pediatrics, Department of Systems Medicine, University of Rome “Tor Vergata”, 00185 Rome, Italy; chiara.rossetti@opbg.net (C.R.); annalisa.deodati@opbg.net (A.D.); 3Department of Neuroscience, Rehabilitation, Ophthalmology, Genetics, Maternal and Child Health (DINOGMI), University of Genoa, 16145 Genova, Italy; gandrea.rotulo@opbg.net; 4Academic Department of Pediatrics (DPUO), Diabetology and Growth Disorders, Bambino Gesù Children’s Hospital, IRCCS, 00165 Rome, Italy; 5Department of Anesthesia and Critical Care, Bambino Gesù Children’s Hospital, IRCCS, 00165 Rome, Italy; roberto.bianchi@opbg.net; 6Neuroradiology Unit, Imaging Department, Bambino Gesù Children’s Hospital, IRCCS, 00165 Rome, Italy; giulia.lucignani@opbg.net (G.L.); daniela.longo@opbg.net (D.L.); 7Department of Neuroscience, Bambino Gesù Children Hospital, IRCCS, 00165 Rome, Italy; massimiliano.valeriani@opbg.net

**Keywords:** severe acute respiratory syndrome coronavirus 2 (SARS-CoV-2), COVID-19 vaccination, multisystem inflammatory syndrome in children (MIS-C), multisystem inflammatory syndrome following SARS-CoV-2 vaccination (MIS-V), mild encephalitis/encephalopathy with reversible splenial lesion (MERS)

## Abstract

The SARS-CoV-2 vaccine roll-out has been successful around the world. However, there are increasing concerns about adverse events. We report two pediatric cases of Multisystem-Inflammatory-Syndrome (MIS-C) with neurological involvement that occurred after SARS-CoV-2 vaccination and unknown recent SARS-CoV-2 infection. Brain magnetic resonance revealed mild-encephalopathy with reversible-splenial-lesion in both cases and complete resolution within 4 weeks. In conclusion, this report aims to describe rare emerging clinical entities that can help pediatricians to make an early diagnosis and to provide appropriate treatment. Multisystem-Inflammatory-Syndromes following COVID-19 vaccination remain rare events. When a history of a recent contact with SARS-CoV-2 is present, a careful evaluation by the clinicians in charge of immunization activities is suggested prior to proceeding with the vaccination.

## 1. Introduction

Multisystem Inflammatory Syndrome (MIS) in children and adolescent (MIS-C) is a condition that usually occurs 2–6 weeks after Severe Acute Respiratory Syndrome Coronavirus 2 (SARS-CoV-2) infection. 

MIS-C is a febrile syndrome with generalized inflammation and multi-organ involvement including abdominal symptoms like pain, vomiting and diarrhea, skin rash and often hypotension and cardiac dysfunction [1].

Besides late inflammatory condition, neurological disorders related to SARS-CoV-2 infection were also described. These include the mild encephalitis/encephalopathy with reversible splenial lesion (MERS) recently reported in children with MIS-C [2,3].

The introduction of vaccines to prevent SARS-CoV-2 infection has been one of the key strategies in the global control of the COVID-19 pandemic.

Adverse effects from COVID-19 vaccination are usually minor, including local pain at injection site, fever, asthenia or general musculoskeletal pain [4].

In particular, neurological adverse events following vaccination are generally mild and transient, although rare major neurological complications, such as cerebral venous sinus thrombosis, Bell’s palsy, acute transverse myelitis, acute disseminated encephalomyelitis, and acute demyelinating polyneuropathy, have been described [5,6]

Recently, a multisystem inflammatory syndrome following SARS-CoV-2 vaccination (MIS-V) both in adults [7] and children [8,9] has been described, raising some concerns in the scientific community which, in conjunction with international regulatory agencies, is actively monitoring this condition [8,9]. MIS-V is a rare entity, and the exact incidence, prevalence and pathophysiology are still unknown. Theories of dysregulation of the T cell responses, cytokine storm and/or hyper-reactivity of the immune system due to a preceding asymptomatic or symptomatic COVID-19 have been suggested [10,11]. However, subclinical SARS-CoV-2 infection around the time of vaccination eventually led to attributing the MIS event to vaccination rather than recent infection. Indeed, most of the MIS-V patients have both a positive test result for SARS-CoV-2 antigen or antibody and a recent family cluster in proximity to a COVID-19 vaccination [7,8,12].

MERS is a clinic-radiological syndrome characterized by a transient mild encephalopathy and a reversible lesion in the splenium of the corpus callosum on MRI. Patients with MERS presented with mild central nervous system symptoms such as consciousness disturbance, seizures and headache, and recovered completely within a month [2]. The pathogenesis of this syndrome is still not completely known. There are several hypotheses, including intra-myelinic edema, axonal damage, hyponatremia, and oxidative stress [3]. To date no post-vaccine MERS cases have been described to our knowledge.

Hereby we described two children who presented MERS and MIS-C after SARS-CoV-2 vaccine and unknown SARS-CoV-2 infection.

### 1.1. Case Report 1

A 14-year-old previously healthy girl presented with history of an episode of catatonia revealed after 2 days of fever and vomit, which arose within 24 h after the first shot of Pfizer/Biontech COVID-19 vaccine. The neurological manifestation resolved in about 7–8 min and was followed by agitation and mental confusion. A month earlier, a familiar cluster of SARS-CoV-2 infection was documented, but the patient’s nasopharyngeal swab for SARS-CoV-2 with real-time polymerase chain reaction (npRT-PCR) resulted negative. Family history resulted positive for autoimmunity: mother affected by Systemic Lupus Erythematosus (SLE). In the past medical history, weight loss associated with skin hyperpigmentation and asthenia had been reported. Physical examination showed skin hyperpigmentation, a mild cutaneous rash and signs of hypotensive shock associated with severe metabolic acidosis and dyselecrolytemia (hyponatriemia, hyperkaliemia and hypoglycemia). The npRT-PCR was negative, but SARS- COV-2 nucleocapsid IgG turned positive. Clinical, laboratory, and imaging findings are shown in Table 1.

Brain magnetic resonance imaging (MRI) showed hyperintensity on T2-weighted images in the splenium of the corpus callosum with restricted diffusion, suggestive of MERS (Figure 1). Electroencephalography (EEG) revealed diffusely slowed background activity.

Diagnostic lumbar puncture was performed, and chemical–physical analysis of cerebrospinal fluid resulted negative as well as microbiological assays.

According with the “Centers for Disease Control and Prevention” (CDC) case definition criteria, the diagnosis of MIS-C was made and intravenous immunoglobulin (IVIG) (2 gr/Kg) was administered. In the suspicion of primary adrenal insufficiency (increased ACTH) intravenous hydrocortisone (100 mg/mq/day) therapy and fluids were promptly started. After 24 h of stress dose of steroid, the patient had significant improvement in mental status. On day 3 of hospitalization, the girl presented respiratory difficulties. Chest X-ray showed right pleural effusion with extension up to the middle of the hemithorax. She started nasal cannula oxygen, human albumin infusion, diuretic and anticoagulant therapy with enoxaparin with gradual improvement of symptoms and complete resolution of pleural effusion after 5 days. The search for anti-adrenal antibodies was subsequently performed with a positive result confirming primary adrenal insufficiency.

She started substitutive therapy with three daily doses of hydrocortisone (15 mg/mq/day) and one daily dose of fludrocortisone (0.1 mg/day). During hospitalization, the symptoms progressively improved, and the patient was discharged from hospital on day 16 after complete recovery. Four weeks after the first image, a brain MRI was repeated, showing complete resolution of the lesion in the splenium of the corpus callosum (Figure 1).

### 1.2. Case Report 2

A previously healthy 9-year-old boy presented with vomiting and diarrhea following 4 days of fever and 10 days after 2nd mRNA-SARS-CoV2 vaccine. In the previous month, he had COVID-19 exposure from a family cluster, but npRT-PCR SARS-CoV-2 resulted negative. Family history resulted positive for autoimmunity, with his mother being affected by lupus nephritis.

At arrival to the emergency department, he was conscious but agitated and hypotensive (85/45 mmHg). He had fever, conjunctivitis, palmar rash, abdominal pain and headache. On admission, npRT-PCR was negative for SARS-CoV-2, but nucleocapsid and spike protein IgG resulted positive. Clinical, laboratory, and imaging findings are shown in Table 1. Echocardiogram showed a left ventricular ejection fraction mildly reduced at 44%. Due to an altered level of consciousness, a brain MRI was performed and showed hyperintensity in the splenium of the corpus callosum with restricted diffusion suggestive of MERS (Figure 2). Evidence of global slowing on the electroencephalography was also detected.

Due to the clinical status and the need to undertake cardiologic supportive therapy, he was admitted to the pediatric intensive care unit (PICU) where IVIG treatment (2 gr/kg) was started in combination with methylprednisolone and enoxaparin. On day 4, considering the persistence of hypotension and the mild improvement of inflammatory markers, treatment with Anakinra was administered at the dosage of 2 mg/kg twice daily.

Subsequently, the patient’s clinical features and vital signs progressively improved and therapy was slowly reduced.

Agitation regressed and echocardiography revealed normalization of ventricular systolic function.

The following brain MRI, performed four weeks after the first imagine, showed no abnormalities (Figure 2), and the patient was discharged on day 21 fully recovered.

## 2. Discussion

To our knowledge, this is the first case series describing MIS-C after SARS-CoV-2 vaccination (MIS-V) and previous unknown SARS-CoV-2 infection, presenting with neurological findings of MERS. Such condition has been recently named as MIS-V due to its clinical presentation, which results quite similar to MIS-C. The affected patients suffer from febrile illness and present multinflammatory symptoms affecting the gastrointestinal, cardiovascular and neurological system associated with elevated markers of inflammation and altered coagulation [13,14].

The pathophysiology and case definition of MIS-V are still under evaluation [13]. The cytokine storm, the dysregulation of immune system as well as host genetics, described in the MIS-C [15], could explain the etiology of MIS-V and the susceptibility to developing symptoms. In our cases, an activation of the immune system due to a preceding SARS-CoV-2 infection (asymptomatic or symptomatic) could be assumed. The subsequent vaccination may result in an additional antigenic stimuli driving an aberrant innate and acquired immune response, resulting in the hyperactivation of proinflammatory cascades. Moreover, in both cases, patients presented a familial history positive for autoimmune diseases (LES). Genome-wide association studies (GWAS) have revealed that many autoimmune conditions are associated with gene variants, which could predispose susceptible individuals to mount an abnormal immunologic response to infectious or environmental exposures [16].

Overall, these hypotheses must be evaluated with caution. Indeed, with the widespread COVID-19 vaccination program, some cases of MIS-C caused by SARS-CoV-2 infections acquired before full vaccine are expected to occur post-vaccination, and will appear to be temporally associated with the vaccine. It is worth highlighting that a detailed medical history focused on individualization of recent familial cluster of COVID-19 and predisposition to autoimmunity could drive the timing of immunization. Indeed, a recent study reported 15 patients including pediatric cases and young adults with earlier diagnoses of MIS-V, but with past or recent SARS-CoV-2 infection revealed by anti-nucleocapsid antibodies [8].

The overlapping neurological findings of MERS observed in both cases is of particular interest. As more patients gain access to COVID-19 vaccines, a wide spectrum of neurological complications is continuously being reported [6]. Several pathogenic mechanisms, like molecular mimicry, direct neurotoxicity, and aberrant immune reactions, have been ascribed to explain these vaccines associated with neurological complications [5]. MERS is a clinic-radiological syndrome characterized by a transient mild encephalopathy and a reversible lesion in the splenium of the corpus callosum on MRI. This entity could be associated with viral infection, but the pathogenesis is still not completely known [5]. It is probably due to a complication secondary to the inflammation and hypercytokinemia in infected subjects without any attributed causative agent in cerebrospinal fluid cultures. Some authors suggest a possible pathogenic role of hyponatremia for MERS [17,18]. In our report, both patients showed hyponatremia associated with MIS-V. Therefore, as suggested by previous case series description [16,17], the combination of marked hypercytokinemia and hyponatremia could be considered to be a possible pathogenetic factor for the development of MERS.

MERS has been recently described in two children with MIS-C without history of COVID-19 vaccination [3]. These children received immunomodulatory drugs with rapid regression of symptoms and complete resolution of the splenial lesion. However, in both our cases, MERS onset was observed after the vaccine administration, albeit with a history of recent Sars-CoV2 infection. Considering the pathogenesis of this specific neurological entity, it is difficult to exclude a causative role of vaccine induced antigen stimulation and subsequent aberrant immune reaction, but a causal association of this adverse event is controversial.

In conclusion, this report and the few previously reported cases emphasize the extreme rarity of MIS after COVID-19 vaccination. Strict surveillance of adverse events related to vaccination is needed because a misattribution of MIS or MERS to vaccination can lead to increased vaccine hesitancy. Therefore, a careful evaluation is required to consider alternative diagnoses and to determine if previous SARS-CoV-2 infection, especially when familial history of autoimmunity is present. In this scenario, serological tests aiming to detect the presence of anti-nucleocapsid antibodies might be informative.

This report has some limitations. It involved a small number of patients, and further study of larger cohorts is necessary to confirm the observations assumed in this report. In particular, additional studies will be needed to investigate SARS-CoV-2 vaccination as a trigger of hyperinflammation and the pathogenic mechanisms underlying transient central nervous system involvement in MIS. Finally, it is of pivotal importance to unravel if many autoimmune conditions are associated with gene variants that could predispose susceptible individuals to mount an abnormal immunological response to SARS-CoV-2 infection or vaccination leading to MIS.

## Figures and Tables

**Figure 1 vaccines-10-01136-f001:**
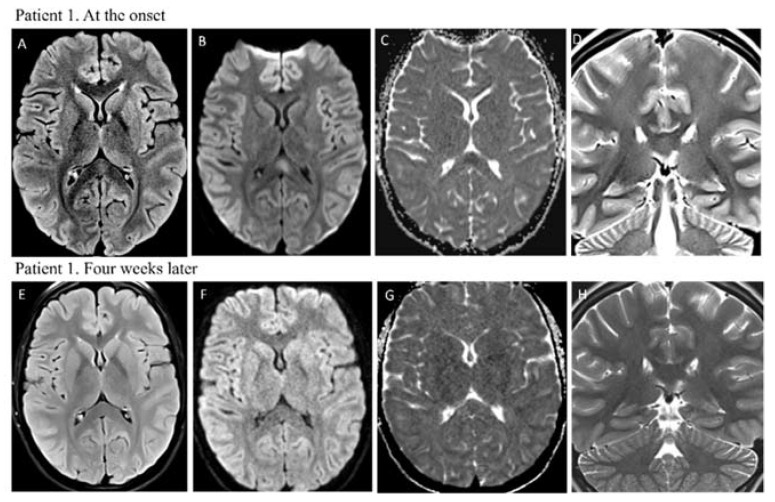
Patient 1 MRI images at the onset showed a hyperintense lesion in the splenium of corpus callosum restricted in DWI/ADC (**B**,**C**), with only low- hyperintensity in FLAIR (**A**) and T2 w images (**D**). Images (**E**–**H**) demonstrated normalization of MRI alteration in the splenial region.

**Figure 2 vaccines-10-01136-f002:**
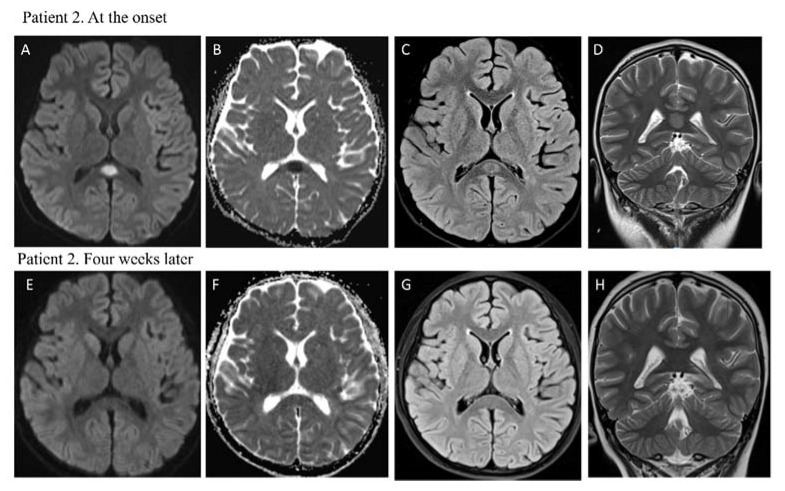
MRI performed at the onset for patient 2 showed focal high signal intensity of splenium of the corpus callosum in diffusion-weighted images (DWI) (**A**) and low signal in ADC map (**B**) in the same lesion. Axial FLAIR (**C**) and coronal T2 w (**D**) images confirmed the presence of a hyperintense splenial lesion. MRI performed one month later revealed complete resolution of the lesion and normal signal intensity of splenium of corpus callosum (**E**–**H**).

**Table 1 vaccines-10-01136-t001:** The table shows the demographic and clinical characteristics of the 2 patients on admission. Laboratory and imaging findings on admission as well as treatments administered during the hospitalization in both patients are also shown.

Title	Patient 1	Patient 2
Age (years)	14	9
Sex	F	M
Initial central nervous system Manifestations	Episode of unresponsiveness to stimuli, catatonia, inability to move followed by agitation and confusion	Agitation and headache, followed by drowsiness
Others symptoms	Fever, vomit	Fever, conjunctivitis, palmar rash, vomiting, diarrhea, abdominal pain
Laboratory findings		
White blood cell (103/uL)	11.87	5.06
Neutrophils (103/uL)	10.03	4.20
Lymphocytes (103/uL)	0.74	0.44
Platelet (103/uL)	314	55
CRP (mg/dL) (N < 0.50)	26.77	16
Procalcitonin (ng/mL)(N: 0–0.5)	4.18	9.96
Ferritin (mg/L)(N: 13–150)	501	1374
Troponin I (pg/mL)(N: <14)	25.7	66.1
Pro-BNP (pg/L)(N: <217)	320	9865
Fibrinogen (mg/dL)(N: 212–433)	700	552
D-dimer (mcg/mL)	8.18	5.35
Albumin (g/dl)	3.1	3.3
Natrium (mEq/L)	127	129
AST (U/L)	46	70
ALT (U/L)	54	32
Creatinine (mg/dl)	1.26	1.17
ACTH (pg/mL)	228	
CSF		
Glucose (mg/dL)	97	
Protein (mg/dL)	20	
Cell count (/mm^3^)	2	
Echocardiogram	Normal ventricular systolic functions	Left ventricular ejection fraction 44%
Chest X-ray	Normal at the admission Day 3: right pleural effusion	Mild accentuation of the broncovascular texture
EEG	global slowing	global slowing
Brain Imaging	MRI: hyperintensity on T2-weighted images in the splenium of the corpus callosum with restricted diffusion	MRI: hyperintensity on T2-weighted images in the splenium of corpus callosum with restricted diffusion
Immune treatment	IVIG Steroid therapy	IVIG Steroid therapy Anakinra
Other treatment	Oxygen	Milrinone
	Enoxaparin	Enoxaparin
	Diuretic	Diuretic

Abbreviations: MRI: Magnetic resonance imaging; IVIG: Intravenous Immunoglobulins.

## Data Availability

Not applicable.

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
