# Peer review of "Two Pediatric Cases of Multisystem Inflammatory Syndrome with Overlapping Neurological Involvement Following SARS-CoV-2 Vaccination and Unknown SARS-CoV2 Infection: The Importance of Pre-Vaccination History"

_vaccines, 2022, doi:10.3390/vaccines10071136_

Round 1

Reviewer 1 Report

This paper described two cases of MIS-V complicated with MERS. Both patients had subclinical COVID-19 infection prior to COVID-19 vaccination. This report would be important for calling attention of clinicians.

Although the author mentioned about the relationship between vaccination and MERS in discussion, hyponatremia was not mentioned as a causative factor for MERS. There are many reports suggesting pathogenic involvement of hyponatremia for MERS. In this report, both patients showed hyponatremia possibly caused by hypercytokinemia. Thus, it would be more appropriate to speculate that MERS developed through the secondary hyponatremia caused by hypercytokinemia associated with MIS-V.

Author Response

This paper described two cases of MIS-V complicated with MERS. Both patients had subclinical COVID-19 infection prior to COVID-19 vaccination. This report would be important for calling attention of clinicians.

Although the author mentioned about the relationship between vaccination and MERS in discussion, hyponatremia was not mentioned as a causative factor for MERS. There are many reports suggesting pathogenic involvement of hyponatremia for MERS. In this report, both patients showed hyponatremia possibly caused by hypercytokinemia. Thus, it would be more appropriate to speculate that MERS developed through the secondary hyponatremia caused by hypercytokinemia associated with MIS-V.

Thank you for your interesting and constructive comment. As you suggested we added in the discussion a speculation on the possible role of hyponatremia associated with hypercytokinemia in the pathogenesis of MERS associated with MIS-V.

Reviewer 2 Report

The article top is novel. But with 2 case reports, it's very skeptical to conclude anything. The author should study more case reports and make sure before they conclude something. 

Author Response

The article topic is novel. But with 2 case reports, it's very skeptical to conclude anything. The author should study more case reports and make sure before they conclude something.

Thank you for the comment. Our report involved a small number of patients, because MERS and Mis-V their self are fortunately rare disorders. However a number of MIS-V cases have been recently reported both in adults ( Belay ED, Godfred Cato S, Rao AK, Abrams J, Wilson WW, Lim S, et al. Multisystem Inflammatory Syndrome in Adults after SARS-CoV-2 infection and COVID-19 vaccination. Clin Infect Dis. 2021 Nov 28:ciab936. Nune A, Iyengar KP, Goddard C, Ahmed AE. Multisystem inflammatory syndrome in an adult following the SARS-CoV-2 vaccine (MIS-V). BMJ Case Rep. 2021 Jul 29;14(7):e243888. doi: 10.1136/bcr-2021-243888) and children  (Karatzios C, Scuccimarri R, Chédeville G, Basfar W, Bullard J, Stein DR. Pediatrics. 2022 May 26. doi: 10.1542/peds.2021-055956; Yousaf AR, Cortese MM, Taylor AW, Broder KR, Oster ME, Wong JM, et al. Reported cases of multisystem inflammatory syndrome in children aged 12–20 years in the USA who received a COVID-19 vaccine, December, 2020, through August, 2021: a surveillance investigation. Lancet Child Adolesc Health (2022)). We are aware that the description of two clinical cases cannot lead to definitive conclusions.  However, the intent of these reports, as the recent one published with 2 cases in Pediatrics, is to describe  rare emerging clinical events that can help Pediatricians to make an early diagnosis, to understand the real frequency of these events and to suggest possible treatments. We add in the conclusion a specific sentence about the limitation of the study. As suggested by the reviewer, an international multicenter collaborative study investigating possible mRNA vaccine-complications in children is currently underway to confirm the conclusions assumed in this study.

Reviewer 3 Report

The Abstract needs thorough revision. I expect the authors to write the need for study and final outcomes in the abstract section.

The introduction needs more information and lack references.

I expect the authors to describe the table properly.

There is no conclusion ?

Author Response

The Abstract needs thorough revision. I expect the authors to write the need for study and final outcomes in the abstract section. The introduction needs more information and lack references.

Thank you for your suggestion. We modified the abstract and the introduction according to your suggestion.We updated the manuscript with last references as requested.

I expect the authors to describe the table properly.

Thank you for your suggestion. We modified the table's description.

There is no conclusion ?

Thank you for your comment. We have expanded the conclusions as you suggested

Round 2

Reviewer 2 Report

I understand that MERS and Mis-V their self are rare disorders, but i am concern regarding the small number of trail to conclude anything. But i like to appreciate the author to work on their conclusion to address my earlier comment. Adding this line "This report has some limitations. It involved a small number of patients and further study of larger cohorts is necessary to confirm the observations assumed in this report." - in the conclusion part will give reader a clear picture that this is an initial or early stage case report. But to confirm the finding more studies are need. So after that consideration, with the new version of the manuscript is ready for acceptance.

Author Response

Thank you for your positive comment.

Best regards